# Evaluation of Antineoplastic Delayed-Type Hypersensitivity Skin Reactions In Vitro

**DOI:** 10.3390/ph15091111

**Published:** 2022-09-06

**Authors:** Inés Roger, Paula Montero, Antonio García, Javier Milara, Pilar Ribera, Jose Alejandro Pérez-Fidalgo, Julio Cortijo

**Affiliations:** 1Biomedical Research Networking Centre on Respiratory Diseases (CIBERES), Health Institute Carlos III, 28029 Madrid, Spain; 2Department of Pharmacology, Faculty of Medicine, University of Valencia, 46010 Valencia, Spain; 3Pharmacy Unit, University Clinic Hospital, 46010 Valencia, Spain; 4Pharmacy Unit, University General Hospital Consortium, 46014 Valencia, Spain; 5Department of Medical Oncology, University Clinic Hospital of Valencia, 46010 Valencia, Spain; 6Biomedical Research Networking Centre on Cancer (CIBERONC), Health Institute Carlos III, 28029 Madrid, Spain; 7INCLIVA Biomedical Research Institute, 46010 Valencia, Spain; 8Research and Teaching Unit, University General Hospital Consortium, 46014 Valencia, Spain

**Keywords:** antineoplastic, DPRA, hCLAT, KeratinoSens^TM^, delayed-type hypersensitivity (DTH)

## Abstract

Delayed-type hypersensitivity (DTH) is caused by a broad number of drugs used in clinic, and antineoplastic drugs show an elevated proportion of DTH, which potentially affects the quality of life of patients. Despite the serious problem and the negative economic impact deriving from market withdrawal of such drugs and high hospitalization costs, nowadays, there are no standard validated methods in vitro or in vivo to evaluate the sensitizing potential of drugs in the preclinical phase. Enhanced predictions in preclinical safety evaluations are really important, and for that reason, the aim of our work is to adapt in vitro DPRA, ARE-Nrf2 luciferase KeratinoSens^TM^, and hCLAT assays for the study of the sensitizing potential of antineoplastic agents grouped by mechanism of action. Our results reveal that the above tests are in vitro techniques able to predict the sensitizing potential of the tested antineoplastics. Moreover, this is the first time that the inhibition of the VEGFR1 pathway has been identified as a potential trigger of DTH.

## 1. Introduction

Delayed-type hypersensitivity (DTH) is caused by a broad number of drugs used in clinics [1], and antineoplastic drugs show an elevated proportion of DTH, which potentially affects the quality of life of patients [2,3,4]. The vague or inconsistent terminology used to describe these reactions may reflect our poor understanding of their pathophysiology, which can vary for different agents. Although severe hypersensitivity reactions are rare, the incidence of mild-to-moderate reactions may be underestimated in the oncology community [5]. Currently, there is no evidence on whether DTH is produced by a particular antineoplastic drug family or mechanism of action, and it has not been formally tested in a DTH predictive assay to evaluate whether there is a chemical drug condition or biological process. In this regard, it would be of potential value to arrange a simple and reliable in vitro test to predict the DTH in both preclinical development and clinical practice.

Skin sensitizers are substances capable of causing DTH, a local skin reaction characterized by redness, swelling, and itching [6]. Reactions vary in severity from self-limiting maculopapular eruptions to life-threatening Stevens–Johnson Syndrome/toxic epidermal necrolysis. Skin sensitization develops in two stages: induction and elicitation. In the induction phase, allergen-specific T cells are generated, a process that typically does not produce clinical symptoms. The elicitation phase and accompanying allergic response occur when a previously sensitized individual is re-exposed to the inducing allergen [7].

DTH reactions to systemically administered drugs are also a safety concern that cannot be revealed by standard toxicity studies [8]. The cells involved and mediators released during the different phases of hypersensitivity reactions can be assessed using in vitro and in vivo tests [9]. However, nowadays, there are no valid in vivo or in vitro methods for assessing the sensitizing potential of a drug during the preclinical phase, despite the important adverse effects induced and directly linked to immune-mediated hypersensitivity and autoimmunity reactions.

The OECD Test Guideline [10,11,12] describes in vitro assays that address mechanisms described under key biological events of the Adverse Outcome Pathway (AOP) [13]. These test methods support the discrimination between skin sensitizers and nonsensitizers by the United Nations Globally Harmonized System of Classification and Labelling of Chemicals [14]. The test methods currently described in this Guideline are direct peptide reactivity assay (DPRA), ARE-Nrf2 luciferase KeratinoSens^TM^ assay, and the human cell line activation test (hCLAT) [15,16,17,18]. The principle of these methods is based on the fact that in the sensitization phase, antigen-specific T cells are generated. Data suggest that pharmaceuticals may share a common mechanism of action with chemical allergens, a fact that supports the possibility to use these in vitro methods for the identification of drugs potentially associated with hypersensitivity reactions [19]. Therefore, this study aims to adapt an in vitro test for DTH to predict the skin sensitizing potential of different antineoplastics. In addition, we investigated whether there is a specific mechanism of action of antineoplastics to produce DTH.

## 2. Results

### 2.1. Taxanes Induce Delayed-Type Hypersensitivity

The sensitizing potential of the antimicrotubule agents, paclitaxel and docetaxel, was analyzed. The molecular initiating event is the covalent binding of electrophilic substances to nucleophilic centers in skin proteins. Paclitaxel and docetaxel were found to have moderate reactivity in DPRA with 85.55 ± 0.74% and 73.55 ± 2.09% cysteine depletion, respectively (Figure 1A,E).

The second key event in DTH is the elicitation of an inflammatory response in the keratinocytes. The proinflammatory potential of VEGF inhibitors was tested in keratinocytes according to the OECD Guidelines [11]. Both drugs were able to elicit inflammatory events in keratinocytes. The Imax of paclitaxel at 12.7 nM was 5.59 ± 1.83 (Figure 1B). At higher concentrations, Imax decreased as cell viability decreased to 31.4 ± 9.2 (Figure 1C). The Imax of docetaxel at 3.7 µM was 2.011 ± 0.47 (Figure 1F). As in the previous case, the higher dose has a lower Imax as viability decreases (Figure 1G).

Finally, the third key event in DHT is the activation of dendritic cells, typically assessed by the expression of specific cell surface markers, chemokines and cytokines [12]. The antineoplastic also caused a significate increase in CD86 and CD54 expression in hCLAT (Figure 1D,H). The feasibility of the results was checked by appropriate controls. 6-methyl coumarin and lactic acid (LA) were used as negative controls, and cinnamic aldehyde (CA) and 1-chloro-2,4-dinitrobenzene (DNCB) was used as positive controls in the DPRA assay. Isopropanol and LA were used as negative controls, and CA and DNCB were used as positive controls in the KeratinoSensTM assay. And in hCLAT, isopropanol and LA were used as negative controls, and DNCB and nickel sulfate (NS) were used as positive controls. The final classification, based on a “2 out of 3”model, predicts that paclitaxel and docetaxel are positive, so they produce skin sensitization (Table 1). 

### 2.2. BCR-ABL, C-KIT, and PDGF Inhibitors Induce Delayed-Type Hypersensitivity

Imatinib, dasatinib, and nilotinib are tyrosine kinase inhibitors that target the BCR-ABL, c-kit, and PDGF (platelet-derived growth factor) receptors. The DPRA reactivity of imatinib was not determined due to its coelution with the cysteine peptide, and the calculation of the percent of peptide depletion was reported as inconclusive. In the case of dasatinib, the cysteine depletion percentages were 17.34 ± 0.39, 32.3 ± 10.03, and 62.13 ± 0.60 for the 0.5 mM, 1 mM, and 2.5 mM concentrations, respectively (Figure 2D). Nilotinib produced high reactivity (99.5 ± 0.59% peptide decrease) at 1mM and moderate reactivity (99.5± 9.29% peptide decrease) at 0.5 mM (Figure 2H). Therefore, they are considered positive for DPRA. In addition, only imatinib was able to elicit inflammatory events in keratinocytes. The Imax of imatinib at 10 µM was 1.85 ± 0.06 (Figure 2A), and the viability, in all concentrations, was higher than 70% (Figure 2B). However, neither of the other two agents are positive as they have an Imax lower than 1.5 (Figure 2E,I). The viability of dasatinib, in all concentrations, was lower than 70%, so the drug is considered cytotoxic (Figure 2F). However, the viability of nilotinib was close to 100% for all concentrations (Figure 2J). Finally, all three drugs cause the activation of dendritic cells. Specifically, imatinib causes an increase in CD86 and CD54 expression (Figure 2C), and dasatinib and nilotinib cause an increase only in CD86 (Figure 2G,K). Therefore, the final classification, based on a “2 out of 3” predicts that imatinib, dasatinib, and nilotinib are positive, so, they produce skin sensitization (Table 1).

### 2.3. Olaparib and Palbociclib Do Not Induce Delayed-Type Hypersensitivity

The sensitizing potential of olaparib and palbociclib was also studied. Olaparib showed low reactivity at 1 mM (with a % cysteine depletion of 21.21%) and moderate reactivity at 2.5 mM (with a % cysteine depletion of 56.89%) (Figure 3A). It was therefore considered positive for the DPRA technique. In the case of palbociclib, it did not produce a percentage of cysteine depletion greater than 13.89% and is thus considered negative (Figure 3E). In the second key event, both olaparib and palbociclib were negative since neither activated the Nrf2 pathway (Imax < 1.5) with a viability greater than 70% (Figure 3B,C,F,G). Olaparib also did not increase the expression of the CD86 and CD54 expression by more than 150% and 200%, respectively (Figure 3D). However, palbociclib increased CD86 expression by more than 150% (Figure 3H). Therefore, the final classification, based on a “2 out of 3”, predicts that olaparib and palbociclib are negative and do not produce skin sensitization (Table 1).

### 2.4. Vegf Inhibitors Induce Delayed-Type Hypersensitivity

The sensitizing potential of vascular endothelial growth factor (VEGF)-coupled tyrosine kinase inhibitors, sunitinib, regorafenib, and sorafenib, as well as the monoclonal antibody against VEGF bevacizumab, were analyzed. Moderate cysteine depletion was noticed at the two highest concentrations of sunitinib and regorafenib (1 mM and 2.5 mM) (Figure 4A,E). Sorafenib showed moderate cysteine depletion at the 2.5 mM concentration and low depletion at 1mM (Figure 4I). Therefore, all of them are considered positive for the DPRA assay. However, by DPRA, it was not possible to evaluate the binding of bevacizumab to cysteine since the DPRA technique is designed for small molecules. However, none of the concentrations of sunitinib, regorafenib, and sorafenib activated the Nrf2 pathway as they had an Imax lower than 1.5 (Figure 4B,F,J). Surely, these compounds did not produce the activation of Nrf2 because at the two highest plasmatic concentrations they decreased cell viability below 50%, thus considered cytotoxic (Figure 4C,G,K). On the other hand, bevacizumab activated the Nrf2 pathway with an Imax of 1.82 ± 0.29, and its viability did not drop below 70% at any of the assayed concentrations (Figure 5A,B).

Finally, concerning the expression of specific cell surface markers, chemokines and cytokines, sunitinib caused an increase of 150% or 200% in relative fluorescence intensity (RFI) of CD86 and CD54, respectively, and therefore was considered positive for both CD96 and CD54 expression (Figure 4D). Regorafenib, sorafenib, and bevacizumab only caused an increase of 150% in RFI of CD86 and therefore were considered positive only for CD86 expression (Figure 4H,L and Figure 5C). Table 1 summarizes the results obtained in the three techniques as well as the final classification based on a “2 out of 3” approach. Out of four VEGF inhibitors tested, all four were positive for h-CLAT and DPRA but not for KeratinoSens^TM^ indicating that they did not elicit a proinflammatory response but did prompt the activation of the third key event.

### 2.5. SiRNA-VEGFR1 Induces Delayed-Type Hypersensitivity

To determine whether there is any relevant mechanism of action in the development of DTH, and because all four VEGFR inhibitors, regardless of their nature, have been positive, we decided to silence the three VEGFR receptors, VEGFR1, VEGFR2, and VEGFR3. Using KeratinoSens^TM^, we observed that siRNA-VEGFR1 activated the Nrf2 pathway, with an Imax greater than 1.5 (Figure 6A) and a viability close to 100% (Figure 6B). Similarly, the only silencing that caused an increase in CD86 expression greater than 150 was siRNA-VEGFR1 (Figure 6C). Therefore, we could conclude that the inhibition of the VEGFR1 pathway is capable of inducing DTH.

## 3. Discussion

Antineoplastic drugs cause delayed hypersensitivity leading to skin reactions ranging from a mild skin rash to life-threatening severe cutaneous adverse reactions, such as Stevens–Johnson Syndrome and toxic epidermal necrolysis [4,20,21]. Patients with Stevens–Johnson Syndrome and toxic epidermal necrolysis present with confluent, dusky erythematous to purpuric plaques, atypical target lesions, bullae, mucositis, and skin detachment. Mortality rates in the general population range from 1–5% for Stevens–Johnson Syndrome and 25–25% for toxic epidermal necrolysis [22]. Despite the serious problem and the negative economic impact deriving from the market withdrawal of such drugs and high hospitalization costs, nowadays, there are no standard validated methods in vitro or in vivo to evaluate the sensitizing potential of drugs in the preclinical phase. For that, the aim of our work was to evaluate the sensitizing potential of different groups of antineoplastic agents grouped by mechanism of action although the results provided in this work cannot differentiate between the grade of severity of these skin hypersensitivity reactions.

In the last decade, incredible progress has been made in the development of a nonanimal test to assess contact hypersensitivity. In vitro methods, such as the myeloid U937 skin sensitization test, hCLAT, and THP-1 activation assay, may be used in the preclinical phase of drug development for hazard identification of the potential to induce DTH. All three methods mentioned are based on the key mechanistic events underlying the awareness process described in the OECD report on AOP for skin sensitization [19]. For all the in vitro methods mentioned, the hypothesis is that traditional drugs or drug metabolites have low molecular weights (<1000 Da), and, as such, they are too small to be “detected” by T cells, and for these reasons, they are unable to spontaneously give an immune reaction. However, similar to chemical sensitizers, they can act as haptens by stably binding to carrier proteins and forming complete high molecular weight immunogenic compounds. Dendritic cells subsequently process this hapten and differentiate into a mature phenotype, characterized by the high expression of costimulatory molecules (CD86 and CD40), adhesion molecules (CD54, CD11a, CD2, and CD58), and release of cytokines (IL-1β, IL-18, and IL-8) [23]. After stimulation, a clone of T cells is produced capable of reacting to the antigen and therefore causing DTH [24].

Starting from the evidence that sensitizing drugs share the same mode of action with chemical sensitizers, we proposed to adapt DPRA, KeratinoSens^TM^, and h-CLAT assays developed for skin sensitizers, also for the identification of the sensitizing potential of different antineoplastic drugs. Our results show that the in vitro assays described in this study could predict such reactions, which would be of potential interest for the preclinical development of numerous drugs.

The literature details that the taxanes, paclitaxel and docetaxel, generate delayed-type hypersensitivity causing severe cutaneous reactions [4,20]. In numerous patients, docetaxel and to lesser extent paclitaxel have been reported to cause chemotherapy-induced Stevens–Johnson Syndrome and toxic epidermal necrolysis [25,26,27,28,29]. In agreement with the literature, our results show that both taxanes are skin sensitizers since all three assays (DPRA, KeratinoSens^TM^, and h-CLAT) are positive. 

Another group of antineoplastics that also generates a lot of skin reactions are tyrosine kinase inhibitors that target the BCR-ABL, c-kit, and PDGF receptors [30,31,32,33]. Toxic epidermal necrolysis and Stevens–Johnson Syndrome have been described as skin effects related to the use of imatinib [31,33]. Fewer side effects have been reported for dasatinib and nilotinib, which can be due to the greater potency and specificity of the drugs. However, despite being more specific, significant skin effects are still being reported [33]. As in clinical practice, in vitro, all three drugs were positive following the prediction model described above.

In addition, we tested two antineoplastics, olaparib and palbociclib, which do not have many cutaneous effects in the clinic. Published clinical trials on palbociclib and olaparib show almost no skin toxicity (NCT01874353, NCT02000622, and NCT01942135). The “2 out of 3” prediction model suggests that both antineoplastics are not skin sensitizers. Therefore, the clinical data correlate with the results obtained in the in vitro techniques. This shows that the in vitro techniques are capable of detecting those compounds that are sensitizing versus those that are not. 

Sunitinib, regorafenib, and sorafenib are small molecule inhibitors of the tyrosine kinase coupled to the VEGFR, as well as bevacizumab, which is a monoclonal antibody against VEGF and usually triggers dermatologic adverse events. There have been reports of typical delayed-type cutaneous hypersensitivity reactions, such as erythema multiforme, Stevens–Johnson Syndrome, and skin rash due to treatment with these antineoplastics, especially with sorafenib [34,35,36,37,38,39,40,41,42]. Diagnoses of these delayed-type cutaneous hypersensitivities were biopsy-confirmed. The above clinical data correlate with the results obtained in the in vitro assays of this work. All three antineoplastics are considered positive since they are positive in at least two of the three assays. However, in KeratinoSens^TM^ assay, sunitinib, regorafenib, and sorafenib, at the two highest plasma concentrations, are not able to activate the Nrf2 pathway because the drugs caused keratinocyte death (the viability was less than 50%). Supporting our results, a recent study identified that sorafenib inhibits mitogen-activated protein kinase MAP3K7 leading to cytotoxicity and keratinocyte injury [43,44]. Adverse skin effects related to DHT after bevacizumab administration are also described in the literature [45]. These findings demonstrate that DHT is independent of the molecule type and VEGF inhibitory pathway. Bevacizumab is considered positive since it is positive in the KeratinoSens^TM^ and hCLAT assays. For DPRA, it was not possible to test it since the technique is designed to evaluate small molecules.

Finally, having confirmed the strong relationship between the patient’s clinical manifestations after antineoplastic administration and the results obtained with DPRA, KeratinoSens^TM^, and h-CLAT, we wondered whether there was any mechanism of action that promoted the sensitization processes. We observed that VEGF inhibitors, with different structures, are sensitizing, so we silenced the different types of VEGF receptors to study whether sensitization is related to a specific signaling pathway. Our results show that the silencing of VEGFR1 results in the activation of the NRF2 pathway and increased expression of CD86; therefore, they are considered positive for KeratinoSens^TM^ and h-CLAT, respectively, and therefore, the inhibition of VEGFR1 is thought to promote DTH. Previous reports have shown that the blockade of the VEGFR3 signaling pathway contributes to the delayed resolution of chronic skin inflammation [46,47]. However, to our knowledge, this is the first study that identifies the VEGFR1 pathway as a participant in delayed hypersensitivity processes. 

Although these techniques were initially designed to determine the sensitizing potential of cosmetics [16,48], we have proven that they can be easily integrated into drug development for the preliminary identification of drug-induced DTH. Other studies have also used these in vitro techniques to assess the sensitizing potential of the systemic administration of nanomaterials and nanotechnology-formulated drug products [49] or to test another drugs, such as clonidine, ofloxacin, procainamide, abacavir, carbamazepine, among others [50,51,52]. However, none of the studies performs the sensitivity analysis by integrating the three in vitro techniques; instead, single tests are used, which may result in a less accurate prediction Therefore, the use of the three techniques could be used for the rapid screening of multiple drugs that cause DTH. However, it is worth mentioning that the use of these techniques to determine the sensitization potential of a drug will only be possible when the mechanism of action of the drug does not interfere with the technique. If the mechanism of action interferes with the in vitro assays, in vivo models, such as the popliteal lymph node assay (PLNA) or its modifications, could be used in research studies for the identification of drugs that may cause DTH [53]. However, the nonanimal test methods (DPRA, KeratinoSens^TM^, and h-CLAT) exhibited good predictions when compared to LLNA data and even better predictions when compared to human data. The ‘2 out of 3’ prediction model achieved accuracies of 90% or 79% when compared to human or LLNA data [18].

Rapid drug screening by these techniques would have several advantages. Among them, in many cases, the skin toxicities described are often related to the efficacy of the treatment; this prediction of the sensitizing potential in vitro could also give us a clue to the efficacy in clinical practice [54,55]. Moreover, due to the difficulties in the differentiation between delayed-type hypersensitivity reactions and type I hypersensitivity, misdiagnosis can occur and result in serious morbidity and mortality due to severe delayed-type hypersensitivity reactions that may occur upon readministration of the drug. Drug market withdrawal is also an important economic issue due to the extremely high costs associated with the development of a drug [56]. Therefore, the use of alternative in vitro techniques described above to detect the sensitization potential during the development phase of a drug would increase safety and possibly reduce the risk of market withdrawal [51].

In conclusion, our results show that DPRA, KeratinoSens^TM^, and h-CLAT are in vitro techniques capable of detecting the potential of antineoplastic to trigger DTH. Moreover, this is the first time that the inhibition of the VEGFR1 pathway has been identified as a potential trigger of DTH.

## 4. Materials and Methods

### 4.1. Chemicals and Reagents

Lactic acid (No. W261114-1KG-K), isopropanol (No. I9516-25ML), cinnamic aldehyde (No. W228613-100G-K), 1-chloro-2,4- dinitrobenzene (DNCB) (No. 237329-10G), 6-methylcoumarin (No. W269905-100G-K), and nickel sulfate (No. N4882-1KG) were purchased from Sigma-Aldrich (St. Louis, MO, USA). Sunitinib (No. S7781), bevacizumab (No. A2006), palbociclib (No. S1116), imatinib (No. S2475), olaparib (No. S1060), dasatinib (No. S1021), nilotinib (No. S1033), bevacizumab (No. A2006), sorafenib (No. S7397), and regorafenib (No. S1178) were obtained from Selleckchem. Paclitaxel (No. HY-B0015) was purchased from MedChemExpress, Monmouth Junction, NJ, USA. Cysteine peptide (Ac-RFAAKAA-COOH) was purchased from Genosphere Biotechnologies, Paris, France. Peptide stock solutions were prepared to a final concentration of 0.667 mM in 100 mM phosphate buffer (pH of 7.5).

### 4.2. Antineoplastic

This study includes drugs that have clear in vivo DTH reported in the literature and other drugs with no reported in vivo DTH. Among them, drugs have different structures, mechanisms of action, and immune reactions. We selected taxanes as positive controls because of the well-known skin hypersensitivity reactions [4,20]. Different antineoplastic drugs that do not interact with the mechanism of action of DHT assays were also selected based on skin hypersensitivity clinical observations, such as vascular endothelial growth factor (VEGF) inhibitors, BCR-ABL, c-KIT, and platelet-derived growth factor (PDGF) inhibitors or antineoplastics with no hypersensitivity clinical observations, such as olaparib and palbociclib. Selected drug concentrations used in in vitro DHT tests were based on human serum concentrations in steady-state after the common dosage described in the summary of product characteristics (European Medicines Agency).

### 4.3. Direct Peptide Reactivity Assay (DPRA)

When the antigen/allergen binds to the peptides of the skin (haptenization), these chemical–peptide adducts interact with keratinocytes and Langerhans cells. These events are evaluated by the DPRA assay [10]. Modified DPRA assay was proposed to address the molecular initiating event of the skin sensitization AOP, namely protein reactivity, by quantifying the reactivity of test chemicals toward model synthetic peptides containing either lysine or cysteine. In our case, we used the cysteine prediction method. Cysteine–peptide depletion values were then used to categorize a substance in one of four classes of reactivity for supporting the discrimination between skin sensitizers and nonsensitizers. The cysteine–peptide reactions were prepared in HPLC glass autosampler vials containing 0.5 mM of the peptide in combination with different concentrations of each test chemical. Calibration standards were prepared from the peptide stock solution at concentrations of 0.534, 0.267, 0.1335, 0.0667, 0.0334, and 0.0167 in 100 mM phosphate buffer (pH of 7.5) with 20% acetonitrile. Reaction controls were performed using DNCB and cinnamic aldehyde (positive controls) or lactic acid and 6-methyl coumarin (negative controls) at 5 mM. A reference control was also performed on the peptide solution at 0.5 mM with the solvent used (acetonitrile). The vials were capped, gently vortexed, and allowed to incubate in the dark at 25 °C for 24 h. Following incubation, the unreacted peptide was quantified by reverse-phase HPLC with UV detection. The chromatographic system comprised a Zorbax SB-C18 column (2.1 mm × 100 mm × 3.5 micron (Agilent Technologies, Santa Clara, CA, USA). UV detection was performed by an SPD-10AV VP UV-vis (SHIMADZU) detector. Separation was achieved using a 0.35 mL/min flow rate with a gradient of mobile phase A (0.1% vol/vol TFA in water) and mobile phase B (0.085% vol/vol TFA in acetonitrile) over a 20 min period. The concentration of peptide was determined in each reaction from the absorbance at 220 nm. The appropriate peak was integrated yielding the peak area corresponding to the individual unreacted peptide in the test reactions, controls, and standards. A linear calibration curve was calculated based on the peptide concentration standards. The calibration curve was accepted as valid if the correlation coefficient (R2) was >0.990. The peptide reactivity was then reported as a percent of peptide depletion, which was determined as the reduction of the peptide concentration in the samples relative to the average concentration of the controls. The samples were analyzed and classified according to the cysteine 1:10 prediction model (threshold 13.89%) [10].

### 4.4. KeratinoSens^TM^ Assay

Antigen/allergen keratinocyte activation generates oxidative stress via Keap1/Nrf2-ARE (antioxidant response element) and can be evaluated by KeratinoSens^TM^ assay [11]. It uses an antioxidant response element (ARE)-coupled luciferase assay for the sensing of specific activation of Nrf2, which is a key regulator of the keratinocyte inflammatory response [57]. KeratinoSens^TM^ assay addresses the second key event of the skin sensitization AOP, namely keratinocyte activation, by assessing, through luciferase induction, the Nrf2-mediated activation of the antioxidant response element (ARE)-dependent genes (OECD, 442D). Skin sensitizers have been reported to induce genes that are regulated by the ARE [58,59,60]. Small electrophilic substances, such as skin sensitizers, can act on the sensor protein Keap1 (Kelch-like ECH-associated protein 1), by, e.g., covalent modification of its cysteine residue, resulting in its dissociation from the transcription factor Nrf2 (nuclear factor-erythroid 2-related factor 2). The dissociated Nrf2 can then activate ARE-dependent genes, such as those coding for phase II detoxifying enzymes [58,61].

The modified KeratinoSens™ assay was performed using KeratinoSens™ cell line, which was obtained from Givaudan (Vernier, Switzerland). The cells were cultured in DMEM supplemented with low glucose, GlutamaxTM, Fetal Calf Serum (FCS) 9.1% (Amimed), and GeneticinTM (500 μg/mL) at 37 °C, in an atmosphere of 5% CO_2_ and 95% humidity. Upon reaching confluency of 80–90%, the cells were seeded in 96-well white plates (10,000 cells/well) for the luciferase assay. Cells were also seeded in clear 96-well plates for the MTT assay. After incubating for 24 h, the medium was replaced with an antibiotic-free medium containing 1% FCS. Before the assay, samples were dissolved in DMSO and diluted in a 1% FCS antibiotic-free medium before adding to the cells. Reaction controls were performed using DNCB (3.9 µM) and cinnamic aldehyde (32 µM) as positive controls and lactic acid (1000 μM) and isopropanol (1000 μM) as negative controls. A reference control was also with the solvent used (DMSO). Cells were treated with test samples or controls for 48 h. At the end of the incubation period, cells were washed with DPBS and incubated with lysis buffer for 20 min at room temperature. Promega firefly luciferase reagent was added, and the luminescence was immediately measured on a Lumistar plate reader (Lumistar Omega, BMG Labtech). An increase in luciferase activity in sample-treated cells was calculated in comparison to DMSO-treated cells (negative control). To determine the cell viability under similar experimental conditions, media were removed from the clear 96-well plate, and cells were incubated with a solution of MTT (0.6 mg/mL in serum-free media) for 4 h at 37 °C. The media were removed, and DMSO was added to each well to dissolve the blue formazan produced by the cells, the color of which was read at 600 nm on a plate reader (Infinite M200, Tecan). [11]. Fold luciferase activity induction was calculated by Equation (1), and the overall maximal fold induction (Imax) was calculated as the average of the individual repetitions.
*Fold induction = (Lsample − Lblank)/(Lsolvent − Lblank)*(1)
where Lsample is the luminescence reading in the test chemical well; Lblank is the luminescence reading in the blank well containing no cells and no treatment; Lsolvent is the average luminescence reading in the wells containing cells and solvent (negative) control.

In order to be considered a positive prediction for KeratinoSens™, the following conditions must be met in the 2 replicates of the experiment or in 2 of the 3 replicates, in case of discrepancy between the first and second replicate:-The Imáx must be equal to or greater than 1.5 and statistically significant compared to the negative control.-Cell viability must be greater than 70% for the lowest concentration of the compound with an Imáx ≥ 1.5-There should be a dose–response increase in luminescence.

### 4.5. Human Cell Line Activation Test (hCLAT)

Dendritic cells recognize the hapten–protein conjugate and migrate to regional lymph nodes through lymphatics. During migration, dendritic cells undergo differentiation and maturation processes, wherein various regulatory cytokines and cell surface maturation biomarkers, such as CD54 and CD86, are expressed [15]. The expression of CD54 and CD86 can be evaluated by hCLAT. Due to the complexity of isolation and high interdonor variability between primary human dendritic cell cultures, the researchers selected the monocyte–macrophage cell lines THP-1 and U-937. They found that these cells possess one of the properties of activated primary dendritic cells; i.e., they express CD86 and/or CD54 in response to allergens. The measured expression levels of CD86 and CD54 cell surface markers were then used for supporting the discrimination between skin sensitizers and nonsanitizers (OECD and 442E).

A modified hCLAT assay was performed in THP-1 cells (obtained from ATCC, Manassas, VA, USA). The cells were cultured in RPMI 1640 media supplemented with 10% fetal bovine serum (FBS) (GE Healthcare Life Sciences, Chicago, IL, USA), 100 U/mL penicillin, and 100 μg/mL streptomycin (Lonza, DE 17-602E, Basel, Switzerland). Test chemicals were dissolved in DMSO. The final DMSO concentration in the assay medium did not exceed 0.2%. For the cell activation assay, THP-1 cells (1 × 106 cells mL per well in 24-well plates) were incubated for 24 h with various concentrations of test samples and controls. Isopropanol (1000 μM) and lactic acid (1000 μM) were employed as negative controls, and DNCB (10 μM) and nickel sulfate (900 μM) were employed as positive controls. Following exposure, the cells were first washed with FACS buffer (Invitrogen, Waltham, MA, USA, 00-4222-26) then resuspended and washed with a blocking buffer containing 0.01% globulins Cohn fraction II/III (G2388-10G, Sigma-Aldrich). Cells were then incubated for 30 min at 4 ºC with the following monoclonal antibodies: APC mouse IgG1 (Dako, X0927), FITC mouse antihuman CD54 (Dako, F7143), and FITC mouse antihuman CD86 (Dako, F7205). The cells were washed and stained with propidium iodide (Sigma-Aldrich, P4170-100MG), and the fluorescence intensity of the viable cells was analyzed using the BD LSR Fortessa X-20. The relative fluorescence intensities (RFIs) of CD86 and CD54 were calculated. If the RFI of CD86 or CD54 was greater than 150 or 200%, respectively, at any dose in at least two experiments, the test chemical was judged as a sensitizer [12]. To accept the hCLAT method, the cell viabilities of medium and solvent/vehicle controls should be higher than 90%, and for the test chemical, the cell viability should be more than 50%. Both acceptance criteria were met in our studies (data not shown).

### 4.6. The “2 out of 3” Prediction Model

Based on the key event of AOP for skin sensitization (OECD, 2012c), we applied an approach that combined two or three test methods. The “2 out of 3” prediction model [18] uses any two congruent results of the three tests (DPRA, KeratinoSens™, and hCLAT) to determine an overall assessment. If at least two of the three assays were positive, the chemical was classified as a skin sensitizer. If at least two of the three assays were negative, the chemical was classified as a nonsensitizer [62].

### 4.7. Real-Time RT-PCR and Sirna Experiments in KeratinoSens^TM^ and THP-1 Cells 

Small interfering RNA (siRNA), including the scrambled siRNA control, was purchased from Ambion (Huntingdon, Cambridge, UK, catalog no. 4390843). FLT1 (*VEGFR1*) gene-targeted siRNA (identification no. ID192, catalog no. AM16708), KDR (*VEGFR2*) gene-targeted siRNA (identification no. ID145034, catalog no. AM16708), and FLT4 (*VEGFR3*) gene-targeted siRNA (identification no. ID145459, catalog no. AM16708) were designed by Ambion. Cells were transfected with siRNA (50 nM) in serum and antibiotic-free medium. The transfection reagent used was lipofectamine-2000 (Invitrogen, Paisley, UK; catalog no. 11668-027) at a final concentration of 2 µg/mL.

Total RNA was isolated using TriPure^®^ Isolation Reagent (Roche, Indianapolis, IN, USA). The integrity of the extracted RNA was confirmed with Bioanalyzer (Agilent, Palo Alto, CA, USA). Reverse transcription was performed in 300 ng of total RNA with a TaqMan reverse transcription reagents kit (Applied Biosystems, Perkin-Elmer Corporation, CA, USA). cDNA was amplified with specific primers and probes predesigned by Applied Biosystems for FLT1 (ID 192), KDR (ID 145034), an FLT4 (ID 145459) in a 7900HT Fast Real-Time PCR System (Applied Biosystems) using Universal Master Mix (Applied Biosystems). Expression of the target gene was expressed as the fold increase or decrease relative to the expression of β-actin as an endogenous control (Applied Biosystems; Hs01060665). The mean value of the replicates for each sample was calculated and expressed as the cycle threshold (Ct). The level of gene expression was then calculated as the difference (ΔCt) between the Ct value of the target gene and the Ct value.

Silencing of the three genes (FLT1, KDR, and FLT4) was confirmed by PCR with the expression reduced by more than 90% (data not shown).

## Figures and Tables

**Figure 1 pharmaceuticals-15-01111-f001:**
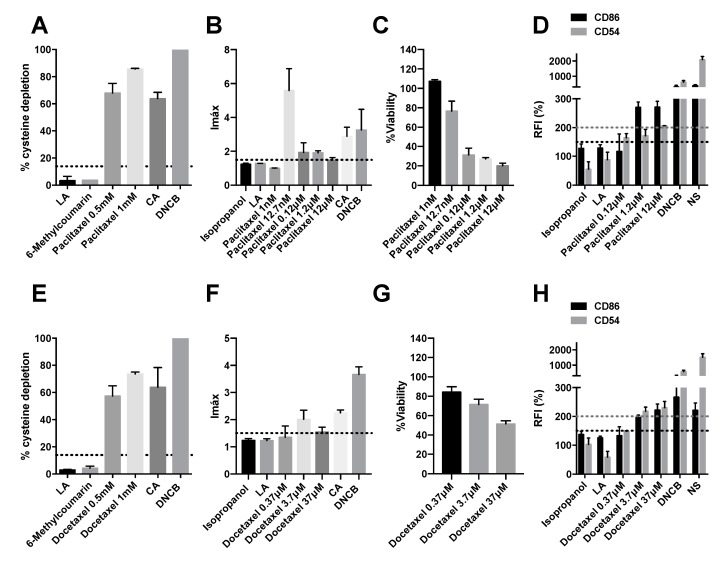
Analysis of delayed-type hypersensitivity of paclitaxel and docetaxel by direct peptide reactivity assay (DPRA), KeratinoSens^TM^ assay, and human cell line activation test (hCLAT). (**A**) DPRA analysis 24 after incubation with pacliaxel. Results are expressed as a percentage of cysteine depletion. (**B**) KeratinoSens^TM^ analysis after 48 h of incubation with paclitaxel. Results are expressed as luciferase activity (Imáx). (**C**) % of viability after 48 h of incubation with paclitaxel. (**D**) Human cell line activation test (hCLAT) analysis after 48 h of incubation with paclitaxel. Results are expressed as relative fluorescence intensities of CD86 and CD54. (**E**) DPRA analysis 24 h after incubation with docetaxel. Results are expressed as a percentage of cysteine depletion. (**F**) KeratinoSens^TM^ analysis after 48 h of incubation with docetaxel. Results are expressed as luciferase activity (Imáx). (**G**) % of viability after 48 h of incubation with docetaxel. (**H**) hCLAT analysis 48 h of incubation with docetaxel. Results are expressed as relative fluorescence intensities of CD86 and CD54. Lactic acid (LA), 6-Methylcoumarin, and isopropanol were used as negative controls. 1-chloro-2,4-dinitrobenzene (DNCB), cinnamic aldehyde (CA), and nickel sulfate (NS) were used as positive controls. Shown are mean ± SEM (*n* = 3). The dashed line depicts the minimum positive stimulation index.

**Figure 2 pharmaceuticals-15-01111-f002:**
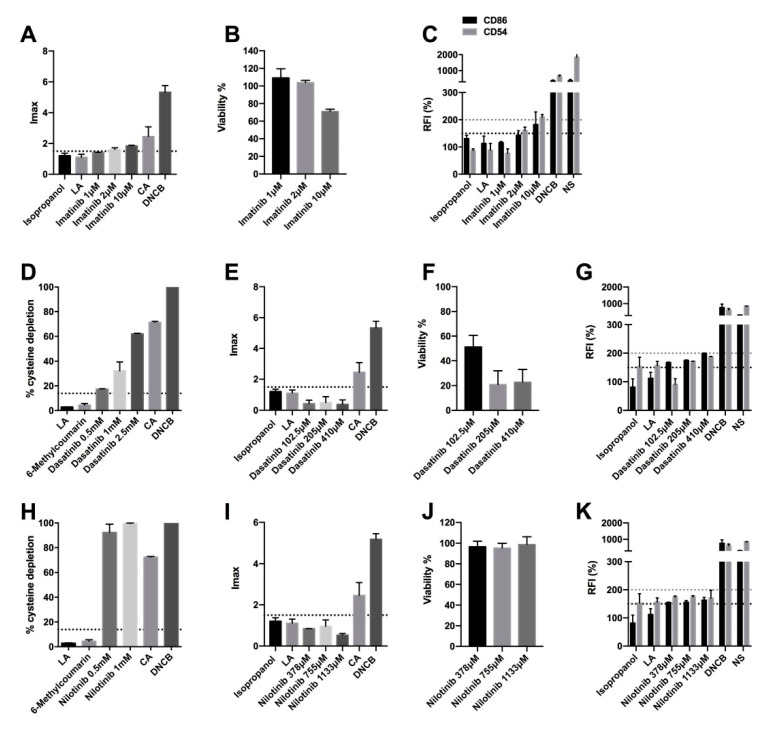
Analysis of delayed-type hypersensitivity of imatinib, dasatinib, and nilotinib by direct peptide reactivity assay (DPRA), KeratinoSens^TM^ assay, and human cell line activation test (hCLAT). (**A**) KeratinoSens^TM^ analysis after 48 h of incubation with imatinib. Results are expressed as luciferase activity (Imáx). (**B**) % of viability after 48 h of incubation with imatinib. (**C**) hCLAT analysis after 48 h of incubation with imatinib. Results are expressed as relative fluorescence intensities of CD86 and CD54. (**D**) DPRA analysis 24 h after incubation with dasatinib. Results are expressed as a percentage of cysteine depletion. (**E**) KeratinoSens^TM^ analysis after 48 h of incubation with dasatinib. Results are expressed as luciferase activity (Imáx). (**F**) % of viability after 48 h of incubation with dasatinib. (**G**) hCLAT analysis after 48 h of incubation with dasatinib. Results are expressed as relative fluorescence intensities of CD86 and CD54. (**H**) DPRA analysis 24 h after incubation with nilotinib. Results are expressed as a percentage of cysteine depletion. (**I**) KeratinoSens^TM^ analysis after 48 h of incubation with nilotinib. Results are expressed as luciferase activity (Imáx). (**J**) % of viability after 48 h of incubation with nilotinib. (**K**) hCLAT analysis after 48 h of incubation with nilotinib. Results are expressed as relative fluorescence intensities of CD86 and CD54. Lactic acid (LA), 6-Methylcoumarin, and isopropanol were used as negative controls. 1-chloro-2,4-dinitrobenzene (DNCB), cinnamic aldehyde (CA), and nickel sulfate (NS) were used as positive controls. Shown are mean ± SEM (*n* = 3). The dashed line depicts the minimum positive stimulation index.

**Figure 3 pharmaceuticals-15-01111-f003:**
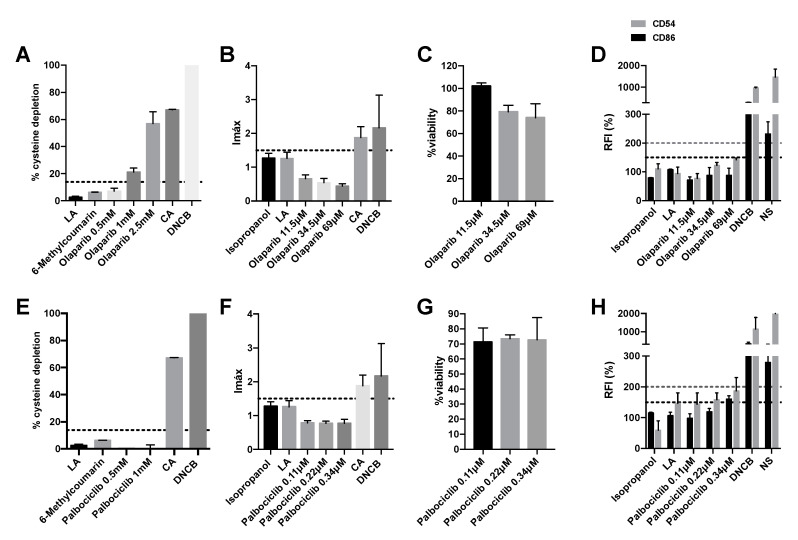
Analysis of delayed-type hypersensitivity of olaparib and palbociclib by direct peptide reactivity assay (DPRA), KeratinoSens^TM^ assay, and human cell line activation test (hCLAT). (**A**) DPRA analysis 24 after incubation with olaparib. Results are expressed as a percentage of cysteine depletion. (**B**) KeratinoSens^TM^ analysis after 48 h of incubation with. Results are expressed as luciferase activity (Imáx). (**C**) % of viability after 48 h of incubation with. (**D**) hCLAT analysis after 48 h of incubation with olaparib. Results are expressed as relative fluorescence intensities of CD86 and CD54. (**E**) DPRA analysis 24 h after incubation with palbociclib 0.5 mM, 1 mM. Results are expressed as a percentage of cysteine depletion. (**F**) KeratinoSens^TM^ analysis after 48 h of incubation with palbociclib. Results are expressed as luciferase activity (Imáx). (**G**) % of viability after 48 h of incubation with palbociclib. (**H**) hCLAT analysis 48 h of incubation with palbociclib. Results are expressed as relative fluorescence intensities of CD86 and CD54. Lactic acid (LA), 6-Methylcoumarin, and isopropanol were used as negative controls. 1-chloro-2,4-dinitrobenzene (DNCB), cinnamic aldehyde (CA), and nickel sulfate (NS) were used as positive controls. Shown are mean ± SEM (*n* = 3). The dashed line depicts the minimum positive stimulation index.

**Figure 4 pharmaceuticals-15-01111-f004:**
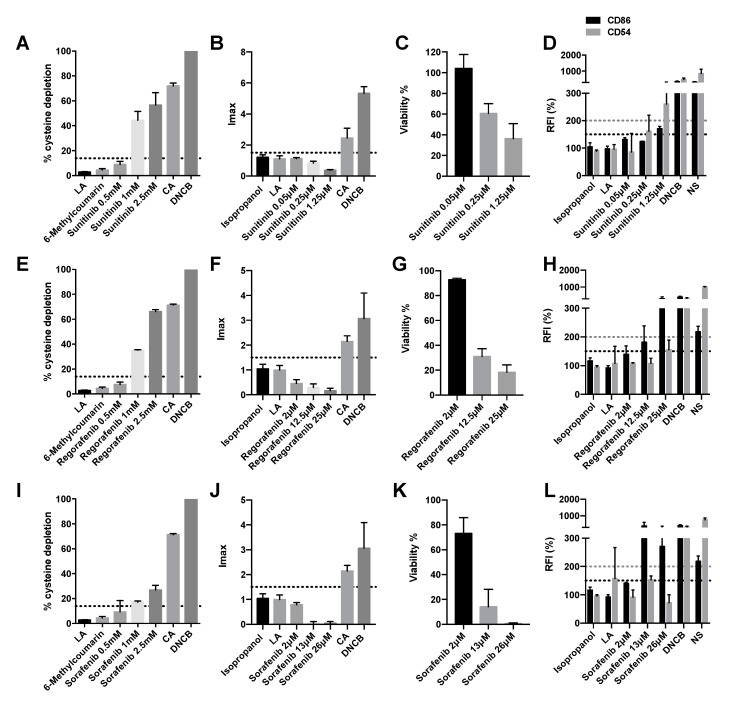
Analysis of delayed-type hypersensitivity of sunitinib, regorafenib, and sorafenib by direct peptide reactivity assay (DPRA), KeratinoSens^TM^ assay, and human cell line activation test (hCLAT). (**A**) DPRA analysis 24 h after incubation with sunitinib. Results are expressed as a percentage of cysteine depletion. (**B**) KeratinoSens^TM^ analysis after 48 h of incubation with sunitinib. Results are expressed as luciferase activity (Imáx). (**C**) % of viability after 48 h of incubation with sunitinib. (**D**) hCLAT analysis 48 h of incubation with sunitinib. Results are expressed as relative fluorescence intensities of CD86 and CD54 (**E**). DPRA analysis 24 h after incubation with regorafenib. Results are expressed as a percentage of cysteine depletion. (**F**) KeratinoSens^TM^ analysis after 48 h of incubation regorafenib. Results are expressed as luciferase activity (Imáx) (**G**) % of viability after 48 h of incubation with regorafenib. (**H**) hCLAT analysis 48 h of incubation with regorafenib. Results are expressed as relative fluorescence intensities of CD86 and CD54. (**I**) DPRA analysis 24 h after incubation with sorafenib. Results are expressed as percentage of cysteine depletion. (**J**) KeratinoSens^TM^ analysis after 48 h of incubation with sorafenib. Results are expressed as luciferase activity (Imáx). (**K**) % of viability after 48 h of incubation with sorafenib. (**L**) hCLAT analysis 48 h of incubation with sorafenib. Results are expressed as relative fluorescence intensities of CD86 and CD54. Lactic acid (LA), 6-Methylcoumarin, and isopropanol were used as negative controls. 1-chloro-2,4-dinitrobenzene (DNCB), cinnamic aldehyde (CA), and nickel sulfate (NS) were used as positive controls. Shown are mean ± SEM (*n* = 3). The dashed line depicts the minimum positive stimulation index.

**Figure 5 pharmaceuticals-15-01111-f005:**
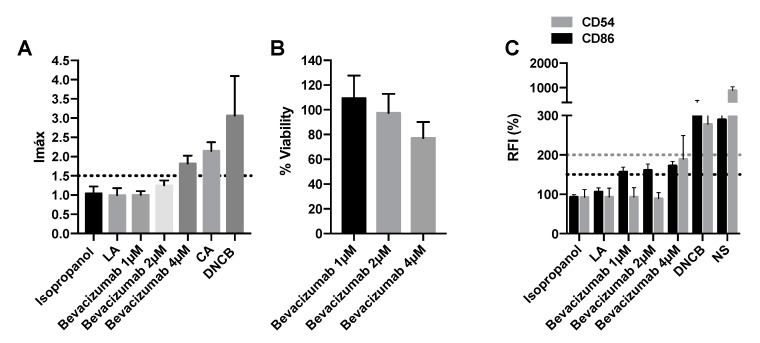
Analysis of delayed-type hypersensitivity of bevacizumab by KeratinoSens^TM^ assay and human cell line activation test (hCLAT). (**A**) KeratinoSens^TM^ analysis after 48 h of incubation with bevacizumab. Results are expressed as luciferase activity (Imáx). (**B**) % of viability after 48 h of incubation with bevacizumab. (**C**) Human cell line activation test (hCLAT) analysis 48 h of incubation with bevacizumab. Results are expressed as relative fluorescence intensities of CD86 and CD54. Lactic acid (LA) and isopropanol were used as negative controls. 1-chloro-2,4-dinitrobenzene (DNCB), cinnamic aldehyde (CA), and nickel sulfate (NS) were used as positive controls. Shown are mean ± SEM (*n* = 3). The dashed line depicts the minimum positive stimulation index.

**Figure 6 pharmaceuticals-15-01111-f006:**
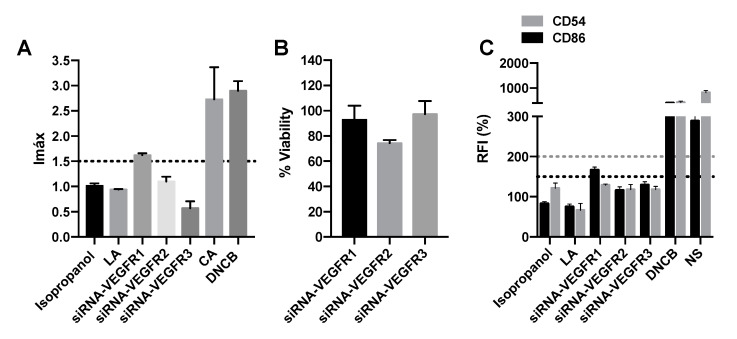
siRNA-VEGF1 mediates delayed-type hypersensitivity. (**A**) Representation of luciferase activity (Imáx) after 48 h of incubation with siRNA-VEGF1, siRNA-VEGF2, and siRNA-VEGF3. (**B**) % of viability after 48 h of incubation with siRNA-VEGF1, siRNA-VEGF2, and siRNA-VEGF3. (**C**) Representation of relative fluorescence intensities of CD86 and CD54 after 48 h of incubation with siRNA-VEGF1, siRNA-VEGF2, and siRNA-VEGF3. Lactic acid (LA) and isopropanol were used as negative controls. 1-chloro-2,4-dinitrobenzene (DNCB), cinnamic aldehyde (CA), and nickel sulfate (NS) were used as positive controls. Shown are mean ± SEM (*n* = 3). The dashed line depicts the minimum positive stimulation index.

**Table 1 pharmaceuticals-15-01111-t001:** Classification results for antineoplastics.

Drug	DPRA	KeratinoSens^TM^	hCLAT	Classification
Sunitinib	+	− *	+	Positive
Regorafenib	+	− *	+	Positive
Sorafenib	+	− *	+	Positive
Bevacizumab		−	+	Positive
Olaparib	+	−	−	Negative
Palbociclib	−	−	+	Negative
Paclitaxel	+	+	+	Positive
Docetaxel	+	+	+	Positive
Imatinib	Inconclusive (Coelution)	+	+	Positive
Nilotinib	+	−	+	Positive
Dasatinib	+	− *	+	Positive

* The viability is below 70%, and therefore the drug is considered cytotoxic.

## Data Availability

The data presented in this study are available in the article.

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
