# Peer review of "Evaluation of Antineoplastic Delayed-Type Hypersensitivity Skin Reactions In Vitro"

_pharmaceuticals, 2022, doi:10.3390/ph15091111_

Round 1

Reviewer 1 Report

The manuscript entitled “Evaluation of antineoplastic delayed-type hypersensitivity skin reaction in vitro” is interesting. I suggest Authors to give more attention to the following remarks point-by-point:

1.     Author badly quotes the literature.

2.     Figure titles are too long.

3.     Why viability is more than 100%.

4.     Table 1 should be early.

I recommend the minor revision of manuscript.

Author Response

The authors would like to express their gratitude for the work carried out by the referees in reviewing this manuscript. The revised manuscript has taken into account all the comments and criticisms raised by the reviewers thus improving, in our opinion, the quality of the revised version.

To ease the review process, in our reply, the reviewer’s comments are pasted just preceding the corresponding replies, as indicated. Also, the main changes introduced in the revised manuscript are highlighted in yellow.

Reviewer 1:

The manuscript entitled “Evaluation of antineoplastic delayed-type hypersensitivity skin reaction in vitro” is interesting. I suggest Authors to give more attention to the following remarks point-by-point:

  1. Author badly quotes the literature. 

Reply

We agree with reviewer 1. We have reviewed the citations in the bibliography and corrected those that were not properly cited.

  1. Figure titles are too long.

Reply

We agree with reviewer 1. We have tried to summarise the figure captions and we have removed the doses as they are detailed in the graphs.

  1. Why viability is more than 100%.

Reply

MTT assay is used to test cell viability. In our case paclitaxel, imatinib, sunitinib, and bevacizumab, at the lowest dose, have a % viability slightly above 100% because these antineoplastics are not toxic at this concentration. According to the literature, MTT assays may present random experimental fluctuation (should be within +/- 10 %), which coincides with our results.

  1. Table 1 should be early.

Reply

The reviewer is right. We have moved the table to section 2.1 as this is where it is first mentioned.

Reviewer 2 Report

Comment for Authors                                    

“Evaluation of antineoplastic delayed-type hypersensitivity skin reactions in vitro

            This paper describes a combination of in vitro assays that can help determine if antineoplastic agents will cause delayed-type hypersensitivity skin reactions.   The authors show good correlation between their results with the three assays described and published clinical results.  This reviewer agrees that these tests would be very helpful in the preclinical phase of drug development to determine if new antineoplastics would cause DTH reactions in the skin.

            While this reviewer found the paper to be scientifically sound, there are

are numerous grammatical errors, typos, and places where the sentence structure needs to be improved (lines 90-94, 157-158, 357) before the manuscript should be published. 

Author Response

The authors would like to express their gratitude for the work carried out by the referees in reviewing this manuscript. The revised manuscript has taken into account all the comments and criticisms raised by the reviewers thus improving, in our opinion, the quality of the revised version.

To ease the review process, in our reply, the reviewer’s comments are pasted just preceding the corresponding replies, as indicated. Also, the main changes introduced in the revised manuscript are highlighted in yellow.

Reviewer 2:

 This paper describes a combination of in vitro assays that can help determine if antineoplastic agents will cause delayed-type hypersensitivity skin reactions.   The authors show good correlation between their results with the three assays described and published clinical results.  This reviewer agrees that these tests would be very helpful in the preclinical phase of drug development to determine if new antineoplastics would cause DTH reactions in the skin.

 While this reviewer found the paper to be scientifically sound, there are numerous grammatical errors, typos, and places where the sentence structure needs to be improved (lines 90-94, 157-158, 357) before the manuscript should be published.  

Reply

The reviewer is right. For this, we have made a revision of the entire manuscript with special emphasis on the lines detailed by the reviewer. All the corrections have been highlighted in yellow in the revised manuscript 
